# Non-Biopsy Serology-Based Diagnosis of Celiac Disease in Adults Is Accurate with Different Commercial Kits and Pre-Test Probabilities

**DOI:** 10.3390/nu12092736

**Published:** 2020-09-08

**Authors:** Venla Ylönen, Katri Lindfors, Marleena Repo, Heini Huhtala, Valma Fuchs, Päivi Saavalainen, Alex Musikka, Kaija Laurila, Katri Kaukinen, Kalle Kurppa

**Affiliations:** 1Celiac Disease Research Centre, Faculty of Medicine and Health Technology, Tampere University, 33520 Tampere, Finland; venla.ylonen@tuni.fi (V.Y.); katri.lindfors@tuni.fi (K.L.); marleena.repo@tuni.fi (M.R.); valma.fuchs@gmail.com (V.F.); alex.musikka@gmail.com (A.M.); katri.kaukinen@tuni.fi (K.K.); 2Tampere Center for Child Health Research, Faculty of Medicine and Health Technology, Tampere University, 33520 Tampere, Finland; kaija.laurila@tuni.fi; 3Department of Pediatrics, Tampere University Hospital, 33520 Tampere, Finland; 4Faculty of Social Sciences, Tampere University, 33520 Tampere, Finland; heini.huhtala@tuni.fi; 5Research Programs Unit, Immunobiology, and Haartman Institute, Department of Medical Genetics, University of Helsinki, 00014 Helsinki, Finland; paivi.saavalainen@helsinki.fi; 6Department of Internal Medicine, Tampere University Hospital, 33520 Tampere, Finland; 7The University Consortium of Seinäjoki, 66320 Seinäjoki, Finland; 8Department of Pediatrics, Seinäjoki Central Hospital, 66320 Seinäjoki, Finland

**Keywords:** celiac disease, anti-transglutaminase 2 antibodies, serology, screening, adults

## Abstract

Non-biopsy diagnosis of celiac disease is possible in children with anti-transglutaminase 2 antibodies (TGA) > 10× the upper limit of normal (ULN) and positive anti-endomysial antibodies (EMA). Similar criteria have been suggested for adults, but evidence with different TGA assays is scarce. We compared the performance of four TGA tests in the diagnosis of celiac disease in cohorts with diverse pre-test probabilities. Serum samples from 836 adults with either clinical suspicion or family risk of celiac disease were tested with four commercial TGA assays, EmA and celiac disease-associated genetics. The diagnosis was set based on duodenal lesion or, in some cases, using special methods. 137 (57%) patients with clinical suspicion and 85 (14%) of those with family risk had celiac disease. Positive predictive value (PPV) for 10×ULN was 100% in each TGA test. The first non-diagnostic investigations were encountered with ULN 1.0×–5.1× in the clinical cohort and 1.3×–4.9× in the family cohort, respectively. Using the assays’ own cut-offs (1×ULN) the PPVs ranged 84–100%. Serology-based diagnosis of celiac disease was accurate in adults using different commercial kits and pre-test probabilities using 10×ULN. The results also suggest that the ULN threshold for biopsy-omitting approach could be lower.

## 1. Introduction

The estimated true prevalence of celiac disease is 1–3% [1,2], emphasizing the importance of efficient and practical diagnostic strategy for this common condition. At the same time, diagnosis of a life-long disease must be based on solid evidence. This has long been achieved by demonstrating characteristic mucosal damage in duodenal biopsy, but such a histology-based approach is invasive and technically challenging [3,4]. Together with the high specificity of modern serological tests, the aforesaid challenges have led to the introduction of new pediatric guidelines for diagnosing celiac disease, enabling a non-biopsy diagnosis in selected children with anti-transglutaminase 2 antibodies (TGA) > 10× the upper limit of normal (ULN) and positive anti-endomysial antibodies (EmA) [5].

It has been suggested that the biopsy-omitting approach could also be extended to adult patients [6,7] and, supporting this, we recently reported a positive predictive value (PPV) of 100% for the serological criteria in adults with variable pre-test probabilities for celiac disease [8]. The results, however, were obtained utilizing only one TGA assay and thus cannot be directly generalized due to a lack of standardization between the commercial tests. In fact, the artificial 10× ULN cutoff and requirement of EmA were introduced mainly to overcome the variation in the diagnostic performance of the TGA-based tests [5]. So far only a few-and exclusively pediatric-studies have directly compared the accuracy of non-biopsy approach for celiac disease with different TGA assays [9].

We investigated this issue by applying four widely used commercial TGA tests in two large and well-defined cohorts of adults with either clinical suspicion (high pre-test probability) or family risk (moderate pre-test probability) of celiac disease.

## 2. Materials and Methods

### 2.1. Patients and Study Design

The study was conducted in the Celiac Disease Research Center, Tampere University and Tampere University Hospital. The patients were collected from among 836 adults, who were further categorized into two sub-cohorts based on assumed pre-test probability for celiac disease (Figure 1). Exclusion criteria were age < 18 years and previous celiac disease diagnosis or otherwise restricted dietary gluten consumption.

The “clinical cohort” with expected high pre-test probability for celiac disease included 239 subjects referred from primary care due to various gastrointestinal and/or extraintestinal symptoms suggestive of celiac disease. They might have been tested previously for celiac disease with serology. All subjects underwent esophagogastroduodenoscopy (EGD) with systematic duodenal sampling (Figure 1), and also participated in research projects that included sampling and storing of sera and whole blood that were subsequently used for testing the studied TGA assays, EmA and celiac disease-associated genetics.

The “family cohort” with presumed moderate pre-test probability for celiac disease consisted of 597 adults, with one or more previously affected relative(s), recruited via newspaper announcements and with the help of the Finnish Celiac Society [10].They underwent sampling and storing of blood for serological and other celiac disease-related measurements similarly to the subjects in the clinical cohort. The option for EGD and biopsies was offered to all subjects with suspicion of celiac disease according to their serology results (Figure 1). The endoscopies were conducted either in Tampere University Hospital or in other local health care units having experience in celiac disease diagnostics.

### 2.2. Ethics

The study protocol and patient enrollment were approved by the regional ethics committee of Pirkanmaa Hospital District (ETL R05183, accepted 6th February 2007). All participants gave written informed consent. The manufacturers of the TGA assays studied had no role in study design, data analysis or interpretation or writing of the manuscript. The study protocol conforms to the ethical guidelines of the 1975 Declaration of Helsinki.

### 2.3. Serological and Genetic Testing

Four different commercial enzyme-linked immunosorbent assays (ELISA) were utilized to test IgA-class TGA, including Celikey (Phadia, Freiburg, Germany), Inova (QUANTA Lite h-tTG, Inova Diagnostics, San Diego, CA, USA), Orgentec (ORG 540A, Orgentec Diagnostika, Mainz, Germany), and Eurospital (Eu-tTG, Trieste, Italy). The cut-offs used for seropositivity were 5 U/mL, 20 U/mL, 10 U/mL and 10 U/mL respectively. The corresponding 10 × ULN were therefore 50 U/mL (Phadia), 200 U/mL (Inova), 100 U/mL (Orgentec), and 100 U/mL (Eurospital), respectively. These were also the upper limit of measuring range for Inova and Eurospital, while those for Phadia and Orgentec were 101 U/mL and 200 U/mL, respectively. All assays studied had passed the appropriate quality controls as requested in the non-biopsy guidelines [5].

EmA were determined in-house with indirect immunofluorescence using human umbilical cord as an antigen [11,12]. A serum dilution 1: ≥5 was considered positive and further diluted until negative or up to 1:4000.

The celiac disease-associated human leucocyte antigen (HLA) genotypes encoding DQ2 and DQ8 molecules on antigen presenting cells were studied from the whole blood samples as described elsewhere [8,10].

### 2.4. Histology

A minimum of four representative forceps biopsies were taken from the duodenum during each EGD. The paraffin-embedded samples were cut, stained with hematoxylin-eosin and studied under a light microscope. Only representative and carefully orientated mucosal sections were included in the histopathological analysis [3]. The majority of celiac disease diagnoses were based on the demonstration of villous atrophy and crypt hyperplasia, equivalent for Marsh 3 lesion [3,13].

In case of milder non-diagnostic lesions, participants were offered additional investigations. Some of the subjects continued on a gluten-containing diet for one year (“gluten challenge”), after which new biopsies were taken and the diagnosis was confirmed if Marsh 3 was present. In a subgroup of patients with non-diagnostic lesions, special diagnostic methods were applied. These included quantitative determination of villous height-crypt depth ratio (VH/CrD) from paraffin sections [3], measurement of mucosal CD3+ (<37 cells/mm) and γδ+ (<4.3 cells/mm) intraepithelial lymphocytes (IEL), and celiac disease-specific IgA deposits from frozen sections [12,14,15]. The subjects started a one-year trial on gluten-free diet (GFD), after which the baseline investigations were repeated, and the diagnosis was set on the basis of positive EmA and IgA deposits and increased CD3+ and γδ+ IELs at baseline and clinical, serological and histological response to the GFD.

Some patients with non-diagnostic duodenal histology had a bullous rash indicative of dermatitis herpetiformis (DH), in which case the diagnosis was confirmed by demonstrating granular IgA deposits in a skin biopsy [16].

### 2.5. Statistics

SPSS^®^ Statistics version 25 (IBM, Armonk, NY, USA) was used for statistical analyses. The data are presented either as number of cases and percentages or as medians with ranges as appropriate. PPV was calculated by dividing the number of true positives (celiac disease) by all test positives (PPV = true positives/[true positives +false positives]). The 95% confidence intervals (CI) for PPV are also given. As a sensitivity analysis, PPV were calculated also considering as true positives only subjects whose celiac disease diagnosis was based on morphological lesion (Marsh 3) in the duodenum (“worst case scenario”). All data were analyzed blinded in collaboration with a medical statistician (H. H.).

## 3. Results

Altogether 125 subjects in the clinical cohort had a diagnostic duodenal lesion in either the primary EGD or after prolonged gluten consumption, and a further 12 received the diagnosis based on special investigations (Figure 1A, Appendix A). Correspondingly, 85 subjects in the family cohort received the diagnosis either directly or after additional investigations (Figure 1B, Appendix A).

The two study cohorts had comparable median ages, while there was a female predominance in the clinical cohort compared with the almost even gender distribution in the family cohort (Table 1). By definition, 100% of the subjects in the family cohort had relative(s) with celiac disease, while the corresponding proportion in the clinical cohort was approximately one-fifth. All participants receiving the diagnosis had HLA DQ2/8 genotype consistent with celiac disease (Table 1).

The overall frequency of seropositivity using manufacturer’s cut-offs for the TGA assays tested ranged from 48.5% to 62.3% in the clinical cohort and from 15.4% to 43.0% in the family cohort. The corresponding numbers in those receiving a celiac disease diagnosis were 84.7–96.6% and 76.5–98.8% (Table 1).

When applying a cut-off 10× ULN, all four TGA assays showed a PPV of 100% in both clinical (95% CIs from 88.0–100% to 92.0–100%) and family (95% CIs from 78.1–100% to 87.0–100%) cohorts (Table 2). With the pre-defined 1× ULN cut-offs the corresponding PPVs ranged in clinical cohort from 83.6% to 100% (95% CIs from 76.0–89.2% to 96.0–100%) and in family cohort from 90.3% to 100% (95% CIs from 82.0–95.2% to 90.7–99.9%), respectively (Table 2). The ULNs calculated by exploiting the highest positive TGA value without celiac disease diagnosis for each assay ranged from 1.0× to 5.1× in the clinical cohort and from 1.3× to 4.9× in the family cohort (Table 3). 

Assuming that only cases with Marsh 3 at any time in the duodenal biopsy or confirmed DH were correctly diagnosed, the PPV for 10× ULN remained 100% in all tests in the family cohort but dropped to 98.1% with QUANTA Lite and to 98.0% with Eurospital in the clinical cohort (Appendix A). The corresponding figures for 1× ULN were 76.9–94.8% in the clinical cohort and 88.2–97.0% in the family cohort (Appendix A). For Celikey and Orgentec, in which the PPV for 10x ULN remained 100% even with “worst case scenario”, the highest values for negative biopsy were 9.6× and 5.3× ULN respectively. 

EmA were positive in 89.8% and 98.8% of those with celiac disease in the clinical and family cohorts, respectively (Table 1). Altogether, EmA was positive in 95.7% of the Celikey, 90.1% of the Orgentec, 78.5% of the Eurospital, and 54.7% of the Inova positive patients; for those who were eventually diagnosed with celiac disease the corresponding figures were 95.6%, 95.0%, 93.6%, and 93.4%, respectively. One subject with TGA >10× ULN in all four tests was EmA negative, as were four subjects with only Eurospital or Inova 10× ULN. All EmA positive participants had HLA DQ2/8.

## 4. Discussion

All four commercial TGA assays tested here demonstrated a PPV of 100% for celiac disease when applying the 10× ULN cutoff as specified by the European Society for Paediatric Gastroenterology Hepatology and Nutrition (ESPGHAN) [5]. The excellent accuracy of the serology-based criteria observed here, if used as recommended, is in line with the majority of recent retrospective and prospective pediatric studies [9,17,18,19,20]. Together with the previous single-assay study by us and some similar reports by other research groups [6,7,21,22,23], these findings provide further evidence that the biopsy-sparing guidelines could also be applied to adult celiac disease patients.

Only a limited number of studies have compared the performance of two or more TGA assays in the serological diagnosis of celiac disease in the same patients; in fact, to the best of our knowledge, such comparisons have been reported only in children [7,9,22,24]. A few previous adult studies have nevertheless utilized two TGA assays in separate cohorts and the result can thus be indirectly compared with our findings [7,22]. Zanini et al. [22] found 5×ULN to be 100% specific for celiac disease with Eurospital and Celikey. However, the ULNs used were 7 U/mL and 16 U/mL for Eurospital (two groups) and 8 U/mL for Celikey, and thus differed from those used here. Efthymakis et al. [7] observed PPVs of 96% and 100% for two tests with 10×ULN, but the assays used were not reported. In children, Werkstetter et al. [9] tested the ESPGHAN criteria meticulously with eight TGA-IgA assays, including Celikey, Inova and Eurospital, and found PPVs of 99.8–100% for >10× ULN. In addition, Rozenberg et al. [24] reported PPVs of >99% for four automated TGA analyzers and 98% for one ELISA kit.

Nevertheless, lower PPVs for non-biopsy approach have also been reported [25,26]. These discrepancies may, at least partially, be attributable to methodological differences, including e.g., variable use of ULN cutoff for a given assay, lack of confirmatory EmA testing, and challenges in applying histology as the reference standard [3,9]. A further explanation for inconclusive histology could be early developing celiac disease presenting with only mild/patchy mucosal changes or extraintestinal form with no apparent intestinal lesion [5,12,16], as also demonstrated here in those who developed diagnostic lesion only after gluten challenge or who had DH. A subgroup of our patients was also diagnosed with a so-called mild enteropathy celiac disease [12,27]. This could be criticized, but they were rigorously investigated with sophisticated diagnostic methods and demonstrated objectively measured treatment response, strongly supporting the presence of celiac disease. Moreover, even if they had all been considered to be non-celiacs, the PPVs for 10× ULN would still have remained excellent.

Of note, performance of the TGA assays was not affected by the assumed pre-test probability for celiac disease, as they worked equally well in both study cohorts. This is in line with our previous single-assay study [8] and dispels the fears that lower pre-test probability for the disease, particularly in screen-detected individuals, would lead to poorer diagnostic accuracy [6,28]. In fact, results of the present and previous studies indicate that serology actually correlates better with the degree of histological lesion than the severity or nature of the symptoms [29,30,31,32]. Likewise, in recent studies the non-biopsy criteria have been equally reliable in symptomatic and asymptomatic children [19,25,33] and ESPGHAN now allows a non-biopsy approach regardless of the clinical presentation [5].

There was also excellent compatibility between TGA values >10× ULN and EmA positivity. This is important, as EmA is considered as the serological reference test that can be utilized to control for the performance variation of the TGA assays [5]. This finding also indicate that this laborious confirmatory step which is not feasible in all centers could be omitted in case of well-validated TGA tests with values >10× ULN. EmA could nevertheless remain useful in borderline cases. In contrast, as also reported in other recent studies [8,9,19], measurement of celiac disease-associated HLA did not provide additional benefits. Accordingly, HLA is no longer required by ESPGHAN [5] although it can be valuable in the differential diagnostics of seronegative duodenal lesion [34].

Depending on the TGA assay, lower than 10× ULN (1.0×–5.1×) also showed 100% PPV for celiac disease, and similar findings have been reported in a few earlier studies [22,35]. This suggests that the proportion of subjects eligible for non-biopsy approach could be increased. The lack of standardization between the TGA tests and their different diagnostic performances nevertheless complicates this issue. As also seen here, some of the assays are clearly intended more as sensitive screening tests whereas others have higher specificity even with low positive values. One possible option might be serial testing with two TGA assays, but at this point the scarcity of evidence makes it challenging to decide optimal combination. EmA confirmation could still be useful in these circumstances, particularly if aiming to lower the required ULN factor or when testing a new TGA combination. On the other hand, the chosen strategy should not be too complicated for routine clinical use. More studies designed to investigate specifically this issue are urgently needed.

### 4.1. Strengths and Weaknesses 

The main strengths of our study were the use of two large and well-defined cohorts of patients with different pre-test probabilities for celiac disease [3,12]. We were also able to exploit sophisticated diagnostic methods in inconclusive cases, although special investigations were not conducted systemically on all participants. There were also limitations. First, EGDs were not centralized to a single hospital, although this should not be a major source of bias as celiac disease diagnostics are harmonized throughout Finland [36]. Second, some of the patients in the clinical cohort had already been tested for TGA before their referral, but significant bias is again unlikely since the results were comparable with those of the previously untested family cohort. Finally, one more potential shortcoming was that we did not control for a possible reduction of the dietary gluten while awaiting the biopsy.

### 4.2. Conclusions

We found the serology-based diagnosis of celiac disease in adults to be accurate with different commercial TGA assays in individuals with variable pre-test probabilities, further supporting the transition towards a less invasive diagnostic approach also after childhood. The results also suggest that, at least in well-validated TGA assays, the specified cut-off for non-biopsy approach could be lowered, but more research on this issue is needed.

## Figures and Tables

**Figure 1 nutrients-12-02736-f001:**
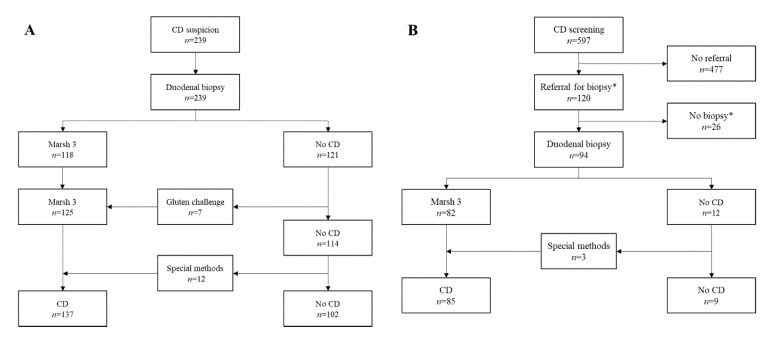
Flowchart of the study. Clinical cohort (**A**) comprises subjects referred from primary care due to suspicion celiac disease (CD). All cases underwent duodenal sampling. Some of the patients with inconclusive histology received the diagnosis in a re-biopsy after one year (“Gluten challenge”). In a subset, the diagnosis was set on the basis of special investigations and clinical, serological, and histological response to gluten-free diet (“Special methods”). The family cohort (**B**) includes subjects with ≥1 relative with celiac disease. Only seropositive subjects were referred for further investigations. *Refusal, self-initiated gluten reduction, and exitus before biopsy.

**Table 1 nutrients-12-02736-t001:** Baseline characteristics and positivity for the serological assays tested in the two study cohorts.

	Clinical Cohort		Family Cohort	
	All Subjects *n* = 239 (%)	Celiac Disease *n* = 137 (%)	All Subjects *n* = 597 (%)	Celiac Disease *n* = 85 (%)
**Baseline Data**				
Age, median (range)	45 (17–83)	45 (18–74)	48 (18–96)	44 (18–80)
Females	72.8	79.6	53.4	50.6
Affected relative	21.1	25.8	100	100
HLA DQ2/DQ8	82.8	100	74.4	100
**TGA positivity**				
Celikey	48.5	84.7	15.4	76.5
Orgentec	51.8	90.8	18.6	88.8
Eurospital	55.5	91.6	23.8	96.3
Inova	62.3	96.6	43.0	98.8
**EmA positivity**	51.5	89.8	19.3	98.8

Data was available on >85% of the subjects in each category. EmA, anti-endomysial antibodies; HLA, human leukocyte antigen; TGA, anti-transglutaminase 2 antibodies.

**Table 2 nutrients-12-02736-t002:** Positive predictive values (PPV) of the four study tests for celiac disease in the clinical and family cohorts.

	≥10× ULN ^a^				≥1× ULN ^a^			
	Positive Subjects (*n*)	Celiac Disease (*n*)	PPV (%)	95% CI (%)	Positive Subjects (*n*)	Celiac Disease (*n*)	PPV (%)	95% CI (%)
**Clinical cohort**								
Celikey	56	56	100	92.0–100	116	116	100	96.0–100
Orgentec	36	36	100	88.0–100	113	108	95.6	89.5–98.4
Eurospital	51	51	100	91.3–100	121	109	90.1	83.0–94.5
Inova	54	54	100	91.7–100	134	112	83.6	76.0–89.2
**Family cohort**								
Celikey	18	18	100	78.1–100	66	65	98.5	90.7–99.9
Orgentec	26	26	100	84.0–100	78	72	92.3	83.4–96.8
Eurospital	33	33	100	87.0–100	84	78	92.9	84.5–97.1
Inova	21	21	100	80.8–100	93	84	90.3	82.0–95.2

^a^ Celikey 5.0 U/mL; Inova 20 U/mL; Orgentec 10 U/mL; Eurospital 10 U/mL. CI, confidence interval; ULN, upper limit of normal.

**Table 3 nutrients-12-02736-t003:** Highest positive anti-transglutaminase 2 antibody value without celiac disease diagnosis for each study assay tested. Above these values the positive predictive value was 100% for all assays.

	Clinical Cohort		Family Cohort	
	Value, U/mL	×ULN ^a^	Value, U/mL	×ULN ^a^
Celikey	4.8	1.0	6.6	1.3
Orgentec	32	3.2	24	2.4
Eurospital	38	3.8	38	3.8
Inova	102	5.1	98	4.9

^a^ Celikey 5.0 U/mL; Inova 20 U/mL; Orgentec 10 U/mL; Eurospital 10 U/mL. ULN, upper limit of normal.

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
