# Peer review of "Non-Biopsy Serology-Based Diagnosis of Celiac Disease in Adults Is Accurate with Different Commercial Kits and Pre-Test Probabilities"

_nutrients, 2020, doi:10.3390/nu12092736_

Round 1

Reviewer 1 Report

This manuscript describes a study to examine the performance of serologic assays in place of utilizing biopsy in the diagnosis of celiac disease. This new study extends an earlier one, which assessed only one anti-TG2 test kit, to now compare multiple anti-TG2 assays in parallel. This is done in fairly large cohorts of adults with either clinical suspicion or family risk of celiac disease, which are representative of what would be seen in the clinic. The manuscript is well-written and the conclusions are well-supported by the data. A few minor issues need to be addressed, as follows.

1) It is stated that  “the corresponding 10x ULN were therefore 50 U/ml (Phadia), 200 U/ml (Inova), 100 U/ml (Orgentec), and 100 U/ml (Eurospital), respectively, which was also the upper limit of measuring range for Inova and Eurospital.” What about the other two assays? Where along the range were the 10x values for Organtec and Phadia?

2) The manuscript uses informal terminology to refer to the various antibodies, such as endomysium and transglutaminase antibodies, etc. Instead, these should be stated as anti-endomysial and anti-transglutaminase 2 (or anti-tissue transglutaminase) antibodies.

3) Lines 145-150. Please indicate which numbers are the 95% CI values. It is confusing as it currently stands.

4) “A further explanation for inconclusive histology could be early developing or extraintestinal celiac disease”. It’s not clear what this sentence aims to communicate. What are “early developing” and “extraintestinal celiac disease”? Please define these clearly, or modify or omit this sentence.

5) It would be helpful to include a sentence on whether the measurement of EMA provided any benefit in cases of TGA values >10x ULN.

Author Response

Reviewer´s comments

Reviewer 1:

The manuscript is well-written and the conclusions are well-supported by the data. A few minor issues need to be addressed, as follows:

1) It is stated that  “the corresponding 10x ULN were therefore 50 U/ml (Phadia), 200 U/ml (Inova), 100 U/ml (Orgentec), and 100 U/ml (Eurospital), respectively, which was also the upper limit of measuring range for Inova and Eurospital.” What about the other two assays? Where along the range were the 10x values for Organtec and Phadia?

Reply: This was a good point and these ranges have now been given in Methods, lines 95-96 as requested.

 2) The manuscript uses informal terminology to refer to the various antibodies, such as endomysium and transglutaminase antibodies, etc. Instead, these should be stated as anti-endomysial and anti-transglutaminase 2 (or anti-tissue transglutaminase) antibodies.

Reply: The used informal terminology has now been corrected.

 3) Lines 145-150. Please indicate which numbers are the 95% CI values. It is confusing as it currently stands.

Reply: These 95% CIs have now been better indicated in lines as kindly suggested; please see Results, lines 150-153. The exact CIs for each antibody are presented in Table 2.

 4) “A further explanation for inconclusive histology could be early developing or extraintestinal celiac disease”. It’s not clear what this sentence aims to communicate. What are “early developing” and “extraintestinal celiac disease”? Please define these clearly, or modify or omit this sentence.

Reply: These vague terms have now been further clarified; please see Discussion, lines 200-201.

 5) It would be helpful to include a sentence on whether the measurement of EMA provided any benefit in cases of TGA values >10x ULN.

Reply: We have now further discussed this important issue as suggested; please see Discussion, lines 218-220.

Reviewer 2 Report

Authors & Editor Summary:

Celiac disease is a serious genetic autoimmune disease that damages the villi of the small intestine and interferes with absorption of nutrients. Approximately 1-3% of the population has celiac disease, but a large percentage of cases are undiagnosed. Current biopsy based approaches are invasive and can be technically challenging. As a result, there is a pressing need to develop reliable serological tests, particularly for adults. In this manuscript, the authors compared four transglutaminase antibody (TGA) tests and one endomysium antibody (EmA) test and compared results against biopsy based diagnosis. The authors identified that the positive predictive value (PPV) was 100% for each TGA test (at 10xULN). The authors have concluded that serology-based diagnosis of celiac disease is possible for adults and should be implemented. This study is very well done and the experiments are both comprehensive and logical. A strength of the paper is the patient population n and comparative results between biopsy and serology. The conclusions are well supported by the findings. I only have only minor comments:

Comments:

  1. In the methods section, can you move up the Ethics section (lines 122-127) to the beginning of the methods. This is an important statement and should be listed first.
  2. In the discussion the authors do an excellent job covering other publications examining TGAs and mentioned that single assay studies appear valid. However, I think it would be valuable for the authors to include a sentence about how many TGAs they think should be implemented into clinical practice to ensure a correct diagnosis (i.e. only 1 TGA, 2 TGAs, a TGA/EmA combo, etc.).

Author Response

 Reviewer 2:

I only have only minor comments:

1) In the methods section, can you move up the Ethics section (lines 122-127) to the beginning of the methods. This is an important statement and should be listed first.

Reply: The place of this section has now been moved into lines 82-87 as kindly suggested.

 2) In the discussion the authors do an excellent job covering other publications examining TGAs and mentioned that single assay studies appear valid. However, I think it would be valuable for the authors to include a sentence about how many TGAs they think should be implemented into clinical practice to ensure a correct diagnosis (i.e. only 1 TGA, 2 TGAs, a TGA/EmA combo, etc.).

Reply: This is indeed a very good question! Although there might not be conclusive answer at this point, but we have now further discussed this important issue as requested; please see Discussion, lines 230-234.